# Developmental Vitamin D Deficiency Produces Behavioral Phenotypes of Relevance to Autism in an Animal Model

**DOI:** 10.3390/nu11051187

**Published:** 2019-05-27

**Authors:** Asad Ali, Svetlina Vasileva, Mia Langguth, Suzanne Alexander, Xiaoying Cui, Andrew Whitehouse, John J. McGrath, Darryl Eyles

**Affiliations:** 1Queensland Brain Institute, The University of Queensland, Brisbane, QLD 4076, Australia; a.ali@uq.edu.au (A.A.); svetlinav@gmail.com (S.V.); suzy.alexander@uq.edu.au (S.A.); x.cui@uq.edu.au (X.C.); j.mcgrath@uq.edu.au (J.J.M.); 2Queensland Centre for Mental Health Research, Brisbane, QLD 4076, Australia; 3Brain and Mind Centre, University of Sydney, Sydney, NSW 2050, Australia; mlan5064@uni.sydney.edu.au; 4Telethon Kids Institute, The University of Western Australia, Perth, WA 6009, Australia; Andrew.Whitehouse@telethonkids.org.au; 5NCRR—National Centre for Register-based Research, Department of Economics and Business Economics, Aarhus University, Aarhus C 8000, Denmark

**Keywords:** Vitamin D, autism spectrum disorder, behavior, brain development, animal model, ultrasonic vocalizations

## Abstract

Emerging evidence suggests that gestational or developmental vitamin D (DVD) deficiency is associated with an increased risk of autism spectrum disorder (ASD). ASD is a neurodevelopmental disorder characterized by impairments in social interaction, lack of verbal and non-verbal communications, stereotyped repetitive behaviors and hyper-activities. There are several other clinical features that are commonly comorbid with ASD, including olfactory impairments, anxiety and delays in motor development. Here we investigate these features in an animal model related to ASD—the DVD-deficient rat. Compared to controls, both DVD-deficient male and female pups show altered ultrasonic vocalizations and stereotyped repetitive behavior. Further, the DVD-deficient animals had delayed motor development and impaired motor control. Adolescent DVD-deficient animals had impaired reciprocal social interaction, while as adults, these animals were hyperactive. The DVD-deficient model is associated with a range of behavioral features of interest to ASD.

## 1. Introduction

Autism spectrum disorder (ASD) is a neurodevelopmental disorder, which typically manifests in the first two to three years of age and persists throughout life [1]. ASD is characterized by difficulties with social interaction, impairments in verbal and non-verbal communications and stereotyped repetitive behaviors [2]. Additionally, there a number of co-morbid features associated with ASD including fine and gross motor delays [3,4] and sensory differences [5] in toddlers and children with ASD.

Global prevalence of ASD is about one in 59 children and adults [6]. It is thought that both genetic and environmental components interact and result in subsequent varied expression of ASD behavioral phenotypes. Recent epidemiological studies indicate that gestational or developmental vitamin D (DVD) deficiency increases the risk of ASD. For example, DVD-deficiency during mid-gestation has been associated with ASD-related traits [7] and ASD diagnosis in six to nine year old children [8]. Children born to vitamin D deficient mothers have a two-fold higher risk of language difficulties compared with mothers with sufficient levels of vitamin D [9]. Lower serum vitamin D level is also associated with impaired motor and physical performance [10,11]. Moreover, a recent systemic review confirmed that higher prenatal vitamin D levels are associated with reduced risk of ASD and other neurodevelopment disorders [12].

The exact mechanism by which DVD-deficiency induces changes in offspring behavior is unclear. However, vitamin D has been shown to play important roles in the regulation of a range of molecular and neurochemical factors that are fundamental for brain development and function [13]. Pre-clinical work in animal models indicates that DVD-deficiency induces several changes in brain systems. Some of these changes include increased brain size, enlarged lateral ventricles and dysregulated dopaminergic system (DA) [14]. Vitamin D treatment during pregnancy has also been shown to rescue ASD-relevant behaviors [15] in an animal models of ASD linked to cerebellar development [16,17]. Taken together, these data support a possible role of vitamin D on developing brain circuits that may be relevant to ASD.

The overarching aim of this study is to investigate the impact of DVD-deficiency in the rat on ASD-relevant behavioral phenotypes. We examined behaviors at a range of ages and maintained a focus on behaviors specific for early post-natal brain development. We repeatedly assessed developmental markers, motor coordination and ultrasonic vocalizations. Older animals were assessed for social interaction, stereotyped repetitive behavior and hyperactivities. Given the nature of some of the early behavioral deficits, we also assessed cerebellum development.

## 2. Materials and Methods

### 2.1. Animals

All procedures performed in this study were approved by the University of Queensland Animal Ethics Committee under the approval number QBI/555/16. Briefly, female outbred Sprague-Dawley (SD) rats were supplied from the Animal Resource Centre, Western Australia. Animals were housed in wired-topped polypropylene cages (510 mm × 330 mm × 190 mm) with woodchip beddings. Animals were kept at a constant temperature of 21 °C ± 2 and 60% humidity on a 12-hour light/dark cycle with ad libitum access to water and food. DVD-deficiency was induced by feeding standard casein AIN93G rodent chow without any added vitamin D (0 IU cholecalciferol) to four weeks old female SD rats. This exposure does not affect maternal serum calcium or phosphorus levels but increases parathyroid hormone in maternal blood [18]. Control female rats were kept on a normal diet (AIN93G) containing 1000 IU of cholecalciferol. Both diets were supplied by Speciality Feeds, Western Australia.

Rats were housed in groups of four. After six weeks on diet, which is sufficient to deplete female SD rats of vitamin D [19], control and vitamin D deficient females were time-mated with control SD sires. Gravid females were housed individually for parturition. Pregnant dams were monitored daily and the day of littering was noted as postnatal day (P) 0. In a departure from many previous studies from our laboratory, here it should be noted that instead of returning dams to a control diet at birth, dams remained on their respective diets until P21. At this point pups were weaned and placed on cereal-based standard rat chow containing normal levels of cholecalciferol. Whole litters were pseudo-randomly divided into three experimental cohorts and underwent behavioral testing as neonates, adolescents and adults (see Figure 1 for details). To test early postnatal behaviors, dams were first removed from the home cage and placed in a clean cage in a holding room. The offspring were then transported to the behavior room in their home-cage and transferred to a pre-heated (37 °C) surgical recovery chamber. A single pup was removed from the cage, behaviorally tested and returned to the home-cage. After testing, the entire litter was transported back to the holding room and the dam returned to the home-cage. The dams were separated from the offspring for no longer than 45 min. For adolescent and adult behavior, rats were transported to the behavior room at least 30 min prior to testing to habituate to the room conditions. All testing was done during the light phase, between the hours of 9 am and 5 pm.

### 2.2. Postnatal Growth and Developmental Milestones

Weight gain was recorded at P3, P7, P30 and P60 for all offspring. Physical development was assessed by recording the postnatal day on which each of the following developmental milestones appeared: Unfolding of the external ear (pinna detachment), fur appearance and eye-opening. Pups were observed daily for the appearance of these milestones.

### 2.3. Behavioral Tests and Procedures

#### 2.3.1. Righting Reflex

The righting reflex is a simple non-invasive test used to assess cerebellum-mediated motor function in neonates. Pups at P3 and P5 were inverted on their backs with all paws facing up in the air and held in position for 5 s and released (Cohort 1, see Figure 1). Each pup was given a scored based on the time it took to return back to four paws after having been placed in a supine position. Pups were scored for a maximum of 30 s [20]. A maximum score of 30 was given if task was not completed in 30 s.

#### 2.3.2. Ultrasonic Vocalizations and Pup Retrieval

Isolation-induced ultrasonic vocalizations (USVs) were recorded at P7 and again at P9 (Cohort 1, Figure 1). USVs were recorded for 3 min in a sound-attenuated chamber. Recordings were obtained using an UltraVox XT system (Noldus Information Technology, The Netherlands), which was capable of recording the full spectrum of sound with a maximum frequency of 160 kHz. Detector outputs were analyzed with UltraVox XT (3.0.80) software (Noldus Information Technology, The Netherlands). Valid calls were detected by applying a specific criterion with a frequency range of 30–70 kHz, minimum amplitude of 40 and a minimum duration of 20 milliseconds with a time gap of 10 milliseconds. 

After recording USVs, dam behavior was assessed by the pup retrieval task. At the start of the test, all pups were shifted to the side of the home cage opposite to the nest and the dam was introduced into the centre of the home cage and video-recorded. Observations were ended when all pups were retrieved, or when 10 min had elapsed [21]. Latencies to retrieve the first, second, third and last pup were recorded.

#### 2.3.3. Negative Geotaxis

Negative geotaxis assesses motor coordination in young rats. P7 pups were placed facing downhill on a 35° inclined surface and on a 40° inclined surface at P10 (Cohort 2, Figure 1). Due to vestibular cues of gravity, pups gradually turn to face up the slope. Pups were timed for completing the 180° upright turn when placed in the head down position. Each pup was given 30 s to complete the task [20,22]. A maximum score of 30 was given if the task was not completed after 30 seconds.

#### 2.3.4. Olfactory Discrimination

Pups were tested for olfactory discrimination at P9 and P11 (Cohort 2, see Figure 1). The apparatus consisted of a clean plastic container (35 cm × 13 cm × 12 cm) with two petri dishes at either end. One petri dish was filled with home cage bedding and the other petri dish had clean bedding. Each pup was placed on a centrally demarcated region with their head away from the experimenter. The latency to enter the home-cage bedding side by crossing the designated line with the front paws or tip of the nose was timed. Each pup was given a maximum of 2 min to complete the task [17,22]. A maximum score of 120 was given if the task was not completed in 2 min.

#### 2.3.5. Social Play Behavior

Adolescent rats between P35 to P40 were tested for social play (rough-and-tumble) behavior (Cohort 3, Figure 1). The apparatus consists of novel chamber (length 52 cm × width 36 cm) containing 2 cm deep wood chip bedding. Prior to testing, rats were habituated to the testing conditions. For the habituation phase of the task (day 1), same-sex littermate pairs from the same cage were placed in the testing chamber for 30 min. On day two, animals were tested for social play behavior by placing same litter-mate pairs together in the testing chamber for 10 min and video recorded. During this session rats were allowed to interact freely. Parameters of social play behaviors such as latency to interact and the frequency of pouncing, pinning and total play duration were analyzed. The data were analyzed on Observer software (Noldus Information Technology, The Netherlands) [23]. As littermate pairs were from the same diet group, they were considered as a single experimental unit for analysis.

#### 2.3.6. Three-Chambered Social Interaction Test

Social interaction (sociability) and preference for social novelty were tested using the three-chambered social interaction test in 80–85 day old rats [24,25]. The apparatus consists of a novel arena (60 cm × 120 cm) composed of three communicating chambers separated by Perspex walls with central openings, which allowed access to all chambers. The task consists of three 10 min phases: Habitation, social preference (for a novel stimulus rat over an object) and social recognition (a preference for a new novel rat over the 1st and now familiar stimulus rat). In the first phase, a column with vertical plastic bars (to allow social contact) was placed in each side chamber and the subject rat was introduced into the arena for phase 1 habituation. After habituation, the second phase of the test is aimed at testing sociability. A stimulus rat (Novel 1), or an object (Lego), was placed within the plastic columns in opposing chambers. The subject rat was then introduced into the central chamber. The subject rat was able to see, smell and physically interact with the stimulus rat. Sociability was defined as the time spent by the subject rat in the chamber containing the stimulus rat compared to the object. In the last phase (social novelty), the novel object was replaced with a second novel rat (Novel 2). This was to assess preference for social novelty in the subject rat, which was defined as the time spent by the subject rat in the chamber with the novel rat (Novel 2) compared to the now familiar stimulus rat. Each stimulus rat was used in two 10 min sessions per day to avoid habituation of the stimulus animals. The data were again analyzed using Ethovision.

#### 2.3.7. Marble Burying

Rats from cohort 3 were tested for stereotyped behavior (repetitive digging) using the marble burying test at P27–31 (adolescence) and again at P56–58 (young adult) using a previously published method [26]. The apparatus consists of a plastic chamber (length 40 cm × width 25 cm) filled with 5 cm deep wood chip bedding. The chamber contained 20 marbles (1.5 cm in diameter), which were evenly spaced about 4 cm apart. The rat was placed in the centre of the chamber. The aim of the task was to assess normal stereotyped digging behavior by assessing how many marbles had been buried to two thirds of their original surface depth.

#### 2.3.8. Elevated Plus Maze

Elevated plus maze (EPM) was used to examine anxiety-like behavior in adult rats at P77–80 (Cohort 3, see Figure 1). The apparatus consists of two open and two closed arms. Each arm was 112 cm long and 10 cm wide. Rats were placed in the centre of the maze facing towards the open arm and allowed to explore the maze for 10 min under dim lighting conditions (lux: Eight open arms and lux: Four closed arms). The percent time spent exploring the open or closed arms and number of entries in each arm were analyzed using Ethovision.

#### 2.3.9. Purkinje Cell Number

At P10, pups from cohort 1 were perfused transcardially with 4% paraformaldehyde in PBS, and brains were post-fixed in the same solution overnight at 4 °C. Following post-fixation, brains were transferred in PBS with 0.05% sodium azide and stored at 4 °C until used. One cerebellar hemisphere of each brain was used for calbindin immunostaining to estimate the total number of Purkinje cells and other hemispheres were processed for Golgi–Cox solution to analyzed dendritic complexity. For calbindin immunostaining, 50 μm thick sections were cut in the parasagittal plane using vibratome (Leica VT1000 S, Leica Microsystems Germany). Free-floating sections were processed for calbindin immunostaining. Antigen retrieval was performed by heating sections at 40 °C for 30 min with 10 mM sodium citrate buffer. Sections were then washed with double distilled water and treated with 0.3% hydrogen peroxide solution for 20 min. Subsequently, sections were treated with blocking solution (3% bovine serum albumin in 0.1% triton and 0.05% sodium azide) to block endogenous rat IgG for 30 min and incubated with 1:5000 dilution of mouse monoclonal anti- calbindin-D-28K antibody (Sigma-Aldrich) for 48 h. Sections were then incubated with secondary antibody (anti-mouse horseradish peroxidase 1:500) for 24 h following three washes with PBS. To visualize immunoreactivity, sections were treated with diaminobenzidine (DAB) for 5–10 mins and rinsed with PBS.

One series of sections (12 sections/series) from 32 different brains (eight brains × two diet groups × two sexes) were imaged using a bright field slide scanner (MetaSystems, Germany) at 20× magnification. Images were segmented into the cell and non-cell pixels with eight segmentation algorithms. Each segmentation algorithm was manually trained using iLastik: A machine learning tool for pixel classification developed by European Molecular Biology Laboratory, Heidelberg Germany. After the training, segmentation algorithms were applied to all the images (Appendix A). This lead to a binary version of original data with all pixels either classified as 1 (cells) or 0 (non-cells) generating cell number. A number of ectopic cells were also observed primarily in lobule 10. These Purkinje cells, which were presumably still migrating at P10 were assessed separately.

#### 2.3.10. Purkinje Cell Dendritic Complexity

One hemisphere from each P10 brain was processed for Golgi–Cox staining (Appendix A). Stored cerebral hemispheres were washed several times with normal saline and placed in Golgi-Cox solution in the dark at room temperature. After 24 h, the Golgi–Cox solution was replaced with the fresh solution and left for a further 10 days at room temperature. Brains were then placed in a 30% sucrose solution at 4 °C for four days. After four days, 150 μm thick sections were cut in the parasagittal plane using vibratome. Sections were mounted on gelatin-coated slides and dried for one day in the dark. To stain, sections were first immersed in 30% ammonia solution for 15 min followed by 1% sodium thiosulfate solution for another 10 min. Between every step, sections were washed with double distilled water for three min. Sections were then processed through dehydration steps (20%, 30%, 50%, 70% and 90% ethanol for 4 min each) and then immersed through three changes of 100% ethanol for 4 min each. Sections were cleared in xylene twice for 3 min and left in fresh xylene overnight. Finally, the sections were cover-slipped using mounting medium (DPX, Merck) and imaged. 

On average, 10 cells were sampled from each hemisphere per animal. To remove selection bias, cells were randomly selected using a multifocal function of ZEISS ZEN Microscope Software. Cells thus selected, were inspected visually before sampling for structural integrity and discarded if any distal or proximal dendrite branches were cut. Images were acquired using Zeiss Axio Imager (Zeiss, Oberkochen, Germany) at 63× objective. All animals were coded and the experimenter blinded to group. Purkinje cell dendrite tracing and sholl analysis for analysis of dendritic complexity were performed using neuron tracing software Neurolucida® (MBF Bioscience, Williston, VT, USA).

#### 2.3.11. Statistical Analysis

Results were analyzed using SPSS (version 25.0) Chicago, IL, USA. In all behavioral tests, statistical significance was measured by multivariate analysis of variance to determine the main effects of maternal diet, sex and maternal diet × sex interactions. Behaviors in which a significant interaction was found were re-analyzed using one-way ANOVA, and where the data were not normally distributed, the non-parametric Mann–Whitney U-test was used to determine statistical significance. The data were expressed as standard error mean (SEM) and the level of statistical significance was defined as *p* < 0.05. For Purkinje cell number, again multivariate analysis of variance was used to determine the main effects of maternal diet, sex and maternal diet × sex interactions. For Purkinje cell dendrite morphology, a two-way ANOVA was conducted as only male animals were used in this experiment.

## 3. Results

### 3.1. Weight Gain and Developmental Milestones

There was no significant effect of DVD-deficiency on weight gain and in both male and female animals when measured at different timepoints (Appendix A). Males were significantly heavier than females at P3 (F_1, 169_ = 4.77, *p* = 0.03), P7 (F_1, 135_ = 9.18, *p* = 0.003), P30 (F_1, 186_ = 17.79, *p* = 0.0001) and P60 (F_1, 95_ = 444.21, *p* = 0.0001). There was no effect of maternal diet on crown-rump length and physical milestones such as ear unfolding, fur appearance and eye-opening (Appendix A).

### 3.2. Righting Reflex

As expected, pup age had a significant effect on the righting reflex. P3 pups were significantly slower to right compared to five day old pups (F_1, 342_ = 54.76, *p* = 0.0001). There was no main effect of sex on the latency to right at P3 (F_1, 169_ = 0.10, *p* = 0.74) and P5 (F_1, 169_ = 1.66, *p* = 0.19). There was also no significant main effect of DVD-deficiency on the latency to right at P3 (F_1, 169_ = 0.05, *p* = 0.81) and P5 (F_1, 169_ = 0.04, *p* = 0.83). However, there was a significant diet × pup sex interaction on the righting reflex at P5 (F_1, 169_ = 4.37, *p* = 0.03). Post-hoc analysis revealed that DVD-deficient male pups had a significantly longer latency to right (U = 900, *p* = 0.01) compared to control males at P5 (Figure 2a).

### 3.3. USVs

Age of pups had a significant effect on calling rates. At P7, pups emitted significantly more calls compared to nine day old pups (F_1, 344_ = 19.65, *p* = 0.0001). There was no significant effect of sex on calling rates at both testing days (P7 (F_1, 170_ = 0.25, *p* = 0.61), P9 (F_1, 170_ = 0.11, *p* = 0.73)).

Calling amplitude was also not significantly different between male and female pups when tested at P7 (F_1, 170_ = 0.31, *p* = 0.57) or P9 (F_1, 170_ = 0.41, *p* = 0.52). Similarly, there was also no sex effect on call duration at P7 (F_1, 169_ = 3.47, *p* = 0.06) and P9 (F_1, 169_ = 0.11, *p* = 0.73).

However, pups born to DVD-deficient dams emitted a greater number of calls compared to control pups at P7 (F_1, 170_ = 25.75, *p* = 0.0001) and P9 (F_1, 170_ = 5.28, *p* = 0.02) (Figure 2b). There was no interaction between sex × maternal diet on calling rates at both testing days (P7 (F_1, 170_ = 0.10, *p* = 0.75), P9 (F_1, 170_ = 1.75, *p* = 0.18)).

Similar to call number, DVD-deficient pups emitted significantly louder (high intensity) calls compared to control pups at both P7 (F_1, 169_ = 23.36, *p* = 0.0001) and P9 (F_1, 169_ = 13.02, *p* = 0.0001; Figure 2c). Again there was no interaction between sex and maternal diet at both testing days.

DVD-deficient pups emitted slightly longer calls at P7 (F_1, 169_ = 5.11, *p* = 0.02) but not at P9 (F_1, 169_ = 1.05, *p* = 0.30; Figure 2d). Again, there was no sex × maternal diet interaction on calling duration at both testing days.

Calling frequency was not altered by pup sex, maternal diet nor was there any sex × maternal diet interaction at P7 and P9.

### 3.4. Pup Retrieval

Figure 3 shows pup retrieval behavior in vitamin D deficient and control dams. At P7, there was no difference among the diet groups on the latency to retrieve the first (F_1, 15_ = 0.01, *p* = 0.91), second (F_1, 15_ = 0.02, *p* = 0.87), third (F_1, 15_ = 0.006, *p* = 0.94) or last pup (F_1, 15_ = 0.007, *p* = 0.94). There was also no main effect of maternal diet on pup retrieval at P9 (First pup (F_1, 15_ = 0.007, *p* = 0.93), second pup (F_1, 15_ = 0.98, *p* = 0.76), third pup (F_1, 15_ = 0.07, *p* = 0.78) and last pup (F_1, 15_ = 0.28, *p* = 0.60)).

### 3.5. Negative Geotaxis

As expected P10 pups were significantly (F_1, 329_ = 11.41, *p* = 0.001) quicker to attain negative geotaxis than P7 pups even though the angle of the incline was increased to make the task more difficult at a later age.

The latency to attain negative geotaxis was not affected by the sex of the pup at both P7 (F_1, 135_ = 0.16, *p* = 0.68) and P10 (F_1, 135_ = 0.24, *p* = 0.62). However, DVD-deficiency had a significant main effect on negative geotaxis (Figure 4a). DVD-deficient pups had significantly longer latency to complete a 180° turn on inclined surface at P7 (F_1, 135_ = 11.36, *p* = 0.001). This effect had washed out by P10 (F_1, 135_ = 3.33, *p* = 0.06). There was also no sex × maternal diet interaction on negative geotaxis at both testing days (P7 (F_1, 135_ = 0.57, *p* = 0.45), P10 (F_1, 135_ = 0.16, *p* = 0.68)).

### 3.6. Olfactory Discrimination

As, expected P11 pups were significantly quicker to reach to the home bedding side compared to P9 pups (F_1, 316_ = 54.09, *p* = 0.0001). Nest seeking behavior mediated by olfactory system was not significantly changed among male and female pups at both P9 (F_1, 196_ = 0.52, *p* = 0.46) and P11 (F_1, 196_ = 0.26, *p* = 0.60). There was also no main effect of DVD-deficiency on olfactory discrimination at both testing days (P9 (F_1, 196_ = 0.54, *p* = 0.46), and P11 (F_1, 196_ = 0.52, *p* = 0.46)). There was also no pup sex and maternal diet interaction on latency to reach home bedding at both testing days (Figure 4b).

### 3.7. Social Play Behavior

Frequency of pouncing was significantly increased (F_1, 94_ = 12.32, *p* = 0.001) in male adolescent rat pairs compared to females (Figure 5a), whereas, the frequency of pinning was not changed (F_1, 94_ = 3.73, *p* = 0.56) between sexes (Figure 5b). Latency to interact (F_1, 94_ = 0.09, *p* = 0.76; Figure 5c) and duration of social exploration/play (F_1, 94_ = 3.10, *p* = 0.08) was also unaltered by sex. Although there was no change observed in either the frequency of pouncing (F_1, 94_ = 0.76, *p* = 0.38), pinning (F_1, 94_ = 0.65, *p* = 0.42) and time spent in social play (F_1, 94_ = 1.49, *p* = 2.22) between diet groups, the latency to initiate interaction was significantly increased (F_1, 94_ = 5.62, *p* = 0.02) in both DVD-deficient male and female rats.

### 3.8. Three-Chambered Social Interaction Assay

In adult animals, in the sociability phase of the three-chambered apparatus, subject rats showed preference for social stimulus (novel rat; time spent investigating novel rat (male = 381.2 ± 149.9, female = 333.4 ± 133.9), over time spent investigating a novel object (male = 113.9 ± 159.9, female = 164.2 ± 143.9); mean ± standard deviation). There were no significant effects of sex on time spent investigating the novel rat (F_1, 48_ = 0.71, *p* = 0.40) or novel object (F_1, 48_ = 1.29, *p* = 0.26; Figure 5d).

There was also no effect of DVD-deficiency on time spent investigating the social stimulus or object as both DVD-deficient and control animals spent same amount of time interacting with novel rat (F_1, 48_ = 0.03, *p* = 0.86) and novel object (F_1, 48_ = 0.05, *p* = 0.87). Moreover, there was no significant interaction between sex and maternal diet on the time spent interacting the novel rat compared to the novel object.

Social novelty could not be explored as the novelty of the 2nd animal (novel 2) was insufficient to promote preferential exploration over the familiar rat (Figure 5e).

### 3.9. Marble Burying

Marble burying was not affected by sex (adolescent (F_1, 187_ = 0.06, *p* = 0.79) and adult (F_1, 95_ = 0.80, *p* = 0.37)). However, DVD-deficiency had a long-term impact on the number of marbles buried (Figure 6a). DVD-deficient animals buried significantly less marbles compared to control animals at both adolescence (F_1, 187_ = 4.25, *p* = 0.04) and adult ages (F_1, 95_ = 7.27, *p* = 0.008).

There was no sex × maternal diet interaction on marble burying at both testing points (adolescent (F_1, 187_ = 0.02, *p* = 0.86), adult (F_1, 95_ = 1.02, *p* = 0.31)).

### 3.10. EPM

Sex had no effect on the number of entries into open (F_1, 94_ = 0.05, *p* = 0.82) or closed arms (F_1, 94_ = 1.20, *p* = 0.27) and time spent in the open (F_1, 94_ = 0.02, *p* = 0.86) or closed arms (F_1, 94_ = 0.5, *p* = 0.81).

Maternal diet had no effect on the number of entries into the open (F_1, 94_ = 0.27, *p* = 0.60) or closed arms (F_1, 94_ = 1.984, *p* = 0.162) and amount of time spent in open (F_1, 94_ = 1.32, *p* = 0.25; Figure 6b) and closed arms (F_1, 94_ = 1.28, *p* = 0.26; Figure 6c). There was also no sex × maternal diet interaction on the number of open or closed arm entries and the time spent in open or closed arms.

Distance travelled in the EPM was also measured. Interestingly, females were hyperactive and travelled significantly more compared to males (F_1, 94_ = 12.63, *p* = 0.001; Figure 6d). More importantly, there was a significant main effect of maternal diet on distance travelled. Our results show that both DVD-deficient males and females travelled significantly further compared to their respective controls (F_1, 94_ = 9.16, *p* = 0.003). There was no maternal diet × sex interaction on the distance travelled in EPM (F_1, 94_ = 4.06, *p* = 0.99).

### 3.11. Estimation of Purkinje Cell Number

Quantification of Purkinje cells revealed that there was no main effect of sex (F_1, 33_ = 0.24, *p* = 0.62) or maternal diet (F_1, 33_ = 0.22, *p* = 0.64) on Purkinje cell numbers across all lobules at P10. No interaction between sex and maternal diet was observed on Purkinje cell numbers across all lobules (F_1, 33_ = 0.03, *p* = 0.86).

At P10, a small number of ectopic Purkinje cells were found in lobule 10. When counted, again there was no effect of sex (F_1, 33_ = 0.11, *p* = 0.74) or diet (F_1, 33_ = 0.13, *p* = 0.71). There was also no significant effect of sex × maternal diet on the number of ectopic cells in lobule 10 (F_1, 33_ = 1.91, *p* = 0.17).

### 3.12. Dendritic Architecture of Purkinje Cells 

Emerging evidence suggest that Purkinje cell dendritic complexity is altered in some animal models that exhibit motor delays. We therefore focused on the effect of DVD-deficiency on dendritic structure of Purkinje cells in these animals as they have shown altered motor and postural control. We examined dendritic architecture in male animals only. Between the two diet groups, no difference was found in total number of primary dendrites (F_1, 145_ = 0.64, *p* = 0.42), total dendritic length (F_1, 145_ = 0.06, *p* = 0.79), number of nodes (F_1, 145_ = 0.29, *p* = 0.59), total number of branch tips (F_1, 145_ = 0.13, *p* = 0.71) and Purkinje cell volume (F_1, 145_ = 2.91, *p* = 0.90). We further calculated dendritic complexity index using a published method [27]. Two-way ANOVA revealed that dendritic complexity was also unaltered in DVD-deficient animals compared with controls (F_1, 145_ = 0.03, *p* = 0.86).

## 4. Discussion

The current study was designed to determine whether DVD-deficient animals exhibit neurodevelopmental difficulties consistent with the ASD behavioral phenotype. A behavioral test battery was used to assess ASD-relevant behavior across a range of developmental stages. The results from this study demonstrate that DVD-deficiency alters many but not all behavioral phenotypes of relevance to ASD in the early postnatal, adolescent and adult offspring.

### 4.1. DVD-Deficiency Induced Delays in Motor Development 

Several studies, including a meta-analysis [28] have consistently shown that children with ASD display difficulties in gross motor function and coordination, which affect their gait. Furthermore, a large population-based longitudinal study has shown that mothers with gestational vitamin D deficiency are more likely to have children with a low score in gross motor skills at 30 months of age [29]. This has also been reported in several animal models of relevance to ASD [17,30,31]. Here we assessed cerebellar-mediated postural and vestibular development by assessing righting reflex and negative geotaxis in young pups. DVD-deficient male pups were slower to self-right at both timepoints with a significant difference at P5 compared with controls. In respect to negative geotaxis, DVD-deficient male and female pups performed worse when compared to controls again at the earlier age tested (P7), but had normal responses at P10. Righting reflex and negative geotaxis tasks are commonly used to assess motor co-ordination in animal models involving perturbations in early development in particular, models of cerebellum dysfunction [32,33].

We therefore investigated whether these abnormal motor functions were caused by delays in cerebellar development. However all aspects of Purkinje cell integrity examined in our study were normal. There may also be non-cerebellar mechanisms operating at earlier time points in DVD-deficient offspring. For example, vestibular dysfunction has been associated with poor balance and postural control in both human [34,35] and animal studies [36]. Accumulating evidence suggests that vitamin D deficiency may be associated with vestibular dysfunction. Vitamin D receptor express in semicircular canal duct which is a major part of the vestibular system [37]. Interestingly, vitamin D receptor mutation in mice leads to neural degeneration in cochlea [38] and sensorineuronal hearing loss [38] suggesting vestibular physiology may be affected in these mice. Moreover, these mice also exhibit impairments in muscular and motor functions [39] further supporting the hypothesis that vitamin D may be linked with vestibular functions. Whether, vestibular physiology is normal in DVD-deficient animals now requires investigation.

### 4.2. DVD-Deficiency Altered Vocal Communication in Pups 

Delays in language development and social communication are prominent features of ASD, and a substantial minority of children with ASD fail to develop functional verbal language by the middle of childhood [40]. Children born to vitamin D deficient mothers have been found to have a two-fold higher risk of language difficulties compared with the mothers with sufficient levels of vitamin D [9].

While language cannot be assessed in rodents, one form of early communication with proven emotional valence is the USVs made by pups when isolated from the nest, presumably as a distress signal [41]. There was a significant main effect of DVD-deficiency on USVs with an increased number and louder call emission in both DVD-deficient male and female pups at both ages tested. These findings are consistent with recent research showing increased USVs in DVD-deficient rats at P12 corroborating our results [42]. These new results contrast with an earlier study by our laboratory which showed no effect of DVD-deficiency on isolation induced USVs in newborn (P0) rat pups [43]. However newborns represent an age that is not typically examined [44] and this earlier study was also potentially compromised by failing to thermally control the environment.

Several genetic and environmental rodent models of ASD with known abnormalities in USVs exhibit decreased [45,46] as well as increased [47,48] calling rates compared with control animals. However, USVs are heavily influenced by strain, post-natal age of assessment and nature of environmental perturbation. For instance, maternal immune activation with high dose of poly(I:C) shows increased USVs [49,50,51]. However offspring prenatally exposed to valproic acid show reduced USV number [52].

The mechanism controlling USVs remains ill defined. One possible mechanism may however involve the neurotransmitter dopamine (DA). DA D_2_ receptor knockout offspring exhibit reduced USVs in response to isolation from the dam [53]. Consistent with this elevating DA levels using amphetamine increased USVs in adult rats and this effect was blocked by halobenzazepine, a DA D1 receptor antagonist [54].

Our laboratory has consistently shown that DVD-deficiency delays DA neuron development in the embryonic brain [55,56]. Whether there is a direct link is a topic for future research.

Alternately, increased calling rates observed in DVD-deficient pups may reflect a heightened state of anxiety in response to isolation. Studies show that when anxiety-inducing drugs are administered to mouse pups, there is an associated increase in the total number of calls emitted [57,58]. Interestingly, we have previously shown that DVD-deficiency increased maternal corticosterone levels [59]. Consistent with this, a recent finding showed down-regulation of genes involved in the inactivation of corticosterone in DVD-deficient mouse placenta [60]. This same group showed that DVD-deficient dams also spent significantly less amount of time licking and grooming their pups [42], which potentially may affect stress responsivity. The exact mechanism behind this robust elevation in USV number and duration in DVD-deficient pups remains unknown.

### 4.3. No Effect of DVD-Deficiency on Pup Retrieval Behavior

USVs are distress signals, which induce maternal exploratory behavior to facilitate retrieval of isolated pups [61]. It is claimed that dams selectively retrieve those pups vocalizing at higher rates [62]. Given the prominent differences in the number of calls among the diet groups, we expected DVD-deficient dams would retrieve their pups sooner [62] however this was not observed. This is in accordance with findings from others who confirm that although DVD-deficiency increases pup vocalizations this does not affect their retrieval [42]. However Yates et al. [42] showed that DVD-deficient dams spent significantly less amount of time licking and grooming their pups in the first postnatal week, however, we did not assessed maternal care behavior in this study.

### 4.4. DVD-Deficiency Altered Marble-Burying/Stereotyped Repetitive Behavior in Adolescent and Adult Rats

Stereotyped repetitive behaviors are considered a core diagnostic feature of ASD in DMS-5 [2]. In patients with ASD repetitive behaviors are characterized by the presence of jumping, spinning, head tilting, nodding arm or hand flapping [63]. In the rodent, one way of assessing such behaviors is marble burying [26,64] as this reflects an ethologically normal stereotyped digging behavior in rodents. DVD-deficient animals buried fewer marbles at both adolescent and adult ages suggesting a long-term persistent effect of DVD-deficiency on this behavior. Reduced marble burying could be a consequence of early motor impairments hinted at by impaired righting reflex and negative geotaxis. However, this seems unlikely as DVD-deficient animals were hyperactive in the EPM and both the EPM and marble-burying were tested in adult rats at similar timepoints.

Most genetic models of relevance to ASD also show decreased marble burying behavior. Such models include Shank mutants [65,66], TPH-2 [67], 5-HT_7_ (serotonin) receptor knock outs [68] and Phosphatase and tensin homolog mutants [69]. However, the situation for environmental models is less clear. In general high dose poly(I:C) increases marble burying [49], however low dose shows reduced rates of burying with no associated motor defect [70]. Importantly vitamin D administration reverses this marble burying defect [70] and has also been shown to reverse stereotyped behaviors in the valproate acid model [15].

The molecular mechanism/s underlying stereotyped digging is poorly understood. The role of the serotonin has been implicated in several stereotyped repetitive behaviors based on pharmacological and genetic studies. For example, the inactivation of the serotonin system by genetic ablation of 5-HT_7_ receptor has been shown to reduce marble burying behavior [68]. Furthermore, a similar reduction in marble burying was observed in wild-type mice injected with a 5-HT_7_ receptor antagonist [68]. Disruption of serotonin synthesis in TPH-2 knockout mice also leads to reduced marble burying in adult mice [67].

DVD-deficiency may modulate stereotyped behaviors possibly due to its developmental effects on serotonin production. We have previously found that serotonin concentration is significantly reduced in DVD-deficient rat brains at birth [71]. Moreover, recent reports have implicated vitamin D in the regulation of the serotonin-synthesizing gene TPH-2 and serotonin metabolism [72,73]. Given that serotonin is important in the establishment of stereotyped behaviors, this could be an important area for further investigation.

### 4.5. DVD-Deficiency Reduced Social Interaction in Adolescent Rats But Not in Adults

Deficiency in social interaction is a core symptom of ASD [2]. The current study is the first to report social play deficiencies in adolescent DVD-deficiency rats. Our data show that, DVD-deficient animals have a longer latency to interact with their playmates compared with controls; however, all other social play measures were normal.

Manipulating DA systems robustly affects social play behavior in rodent [74,75,76,77]. Therefore the well described alterations in DA systems in our DVD-deficient animals [55,56,71,78,79] may again be relevant here.

Our findings of normal social approach in adult offspring using the three-chamber apparatus contrasts with findings from others [42]. It may be that the extended handling of our animals for marble burying and social play prior to assessment in the three-chambered social interaction test may have compromised this measure. We have shown previously that handling can abolish other DVD-deficiency-induced behaviors [59]. Therefore, further characterization of such behavior should be carried out without prior handling.

### 4.6. DVD-Deficiency Induced Hyperlocomotion

DVD-deficient animals travelled significantly further in the EPM, but they did not show alterations in the time spent in open or closed arms. This suggests that DVD-deficient animals had no anxiety-like phenotype but were hyperactive. These results are consistent with earlier studies displaying hyper-locomotion in DVD-deficient animals [59,80,81,82], and is also concordant with our well-described abnormalities in developing DA systems in this model.

## 5. Conclusions

Our results identify broad-spectrum behavioral abnormalities consistent with several features of ASD in DVD-deficient animals. These findings support the use of the DVD-deficient rodent to understand how epidemiologically-informed risk-factors for ASD can change brain function. Consistent with evidence that ASD is a disorder of early brain development, certain behavioral abnormalities were detectable at early neonatal ages. As discussed above, several pathogenic mechanisms are thought to contribute to the increased risk of ASD induced by DVD-deficiency. Most of these mechanism share common pathways with other environmental factors implicated in ASD risk, including maternal immune activation and prenatal exposure to valproic acid. Importantly, behavioral phenotypes in these models have recently been shown to be reversed by vitamin D administration [15,70], suggesting vitamin D may tap into a convergent mechanism behind multiple adverse environmental factors implicated in ASD. Given the emerging epidemiological evidence supporting a role for vitamin D deficiency in neurodevelopmental disorders, we now require further studies to understand the neurobiology behind this constellation of interesting early phenotypes. Moreover, the association between DVD-deficiency and ASD may have important implications for public health.

## Figures and Tables

**Figure 1 nutrients-11-01187-f001:**
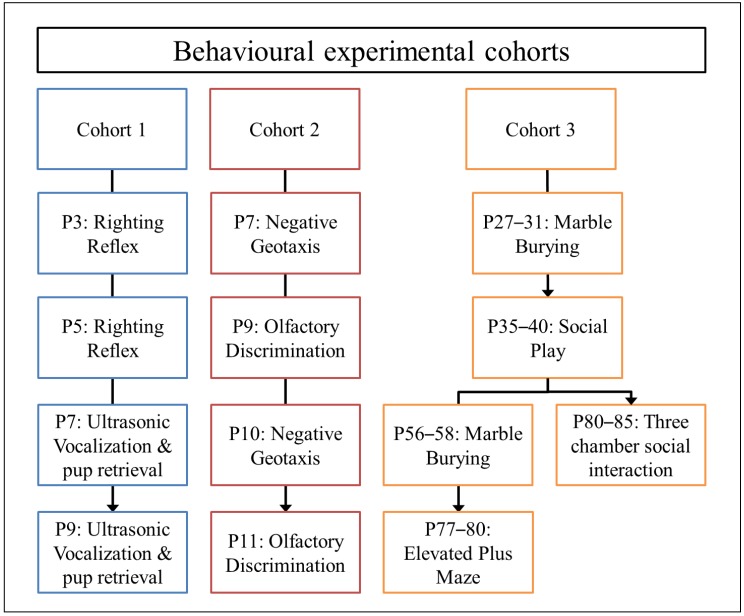
Three separate cohorts were used to assess behaviors in neonates, adolescent and adult rats. Animals were repeated tested at different ages to both assess developmental trajectories and long-term behavioral changes, P = postnatal day.

**Figure 2 nutrients-11-01187-f002:**
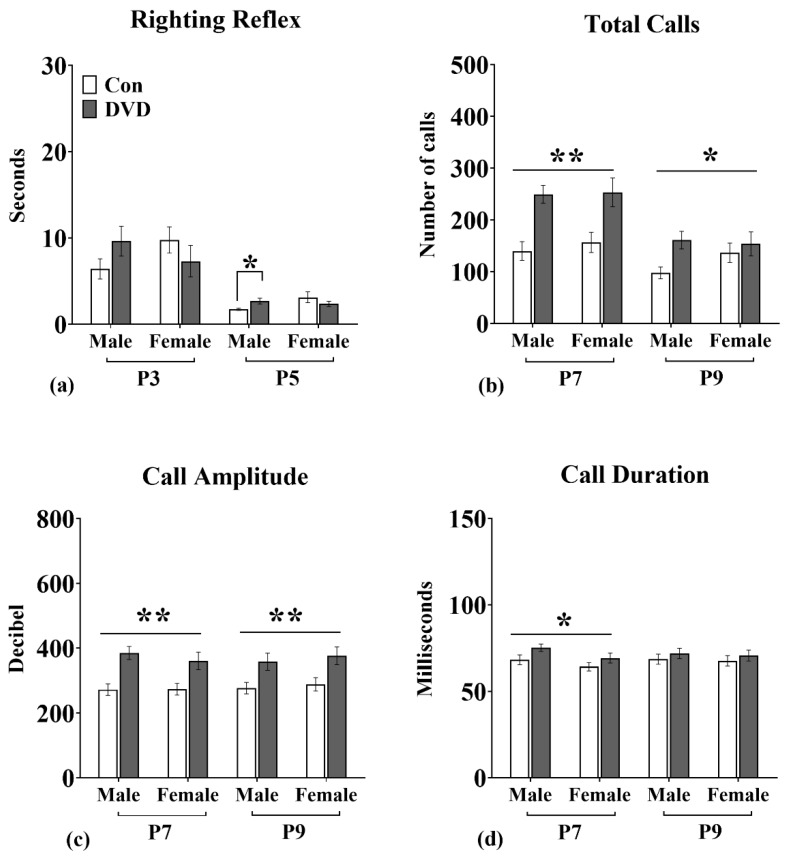
(**a**) The righting reflex was tested at P3 and P5. There was no effect of developmental vitamin D (DVD)-deficiency on the righting reflex at P3. However, DVD-deficient males showed a delayed latency to right at P5 with no difference in female pups. DVD males *n* = 49, control males *n* = 48, DVD females *n* = 29 and control females *n* = 43. Data shown is mean time taken by pups to right in seconds. Ultrasonic vocalizations were recorded at P7 and P9. (**b**) Both DVD-deficient male and female pups emitted significantly greater number and (**c**) louder calls at P7 and P9 compared with control pups. (**d**) DVD-deficient pups also emitted long calls at P7 with no difference in call duration at P9. DVD males *n* = 49, control males *n* = 48, DVD females *n* = 29 and control females *n* = 43. Error bars show SEM, * *p* < 0.05 and ** *p* < 0.0001, P = postnatal day.

**Figure 3 nutrients-11-01187-f003:**
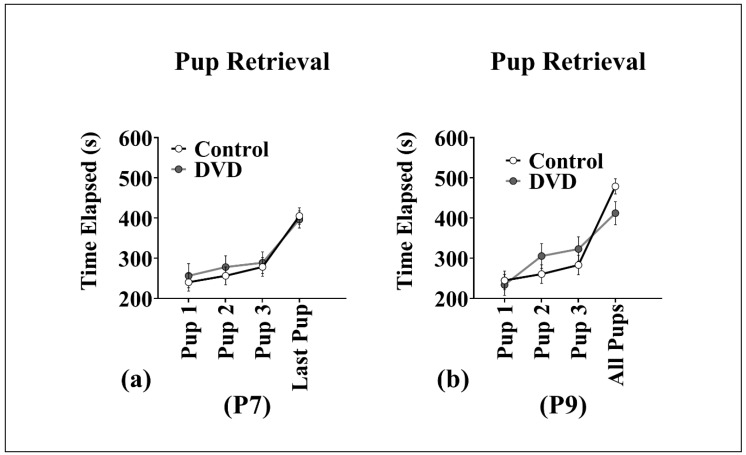
Pup retrieval assay (Cohort 1). Pup retrieval behavior in vitamin D deficient and control dams at P7 and P9. (**a**) There was no effect of vitamin D deficient diet on the dam’s latency to retrieve pups at P7 and (**b**) P9. Data reported as latency to retrieve the first, second, third and last pup in seconds. DVD-deficient dams *n* = 7, control dams *n* = 8 and error bars show SEM, P = postnatal day.

**Figure 4 nutrients-11-01187-f004:**
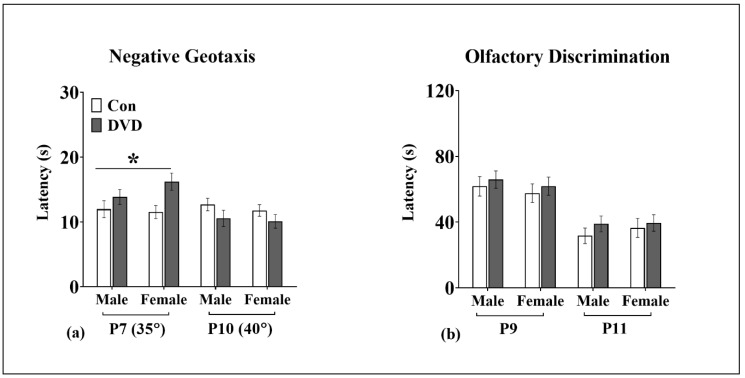
Negative geotaxis and olfactory discrimination (Cohort 2). (**a**) Negative geotaxis was tested P7 and P10. Both DVD-deficient male and female pups had a delayed latency to attain negative geotaxis compared to controls at P7 with no difference among the diet groups at P10. DVD males *n* = 42, control males *n* = 29, DVD females *n* = 34 and control females *n* = 30. Data shown is mean time taken by pups to complete a 180° turn in seconds. (**b**) Olfactory discrimination was tested at P9 and P11. There was no effect of DVD-deficiency on olfactory discrimination in both male and female pups at both testing days. DVD males *n* = 50, control males *n* = 44, DVD females *n* = 63 and control females *n* = 43. Data shown is mean latency to reach the home bedding side measured in seconds. Error bars show SEM, * *p* < 0.05, P = postnatal day.

**Figure 5 nutrients-11-01187-f005:**
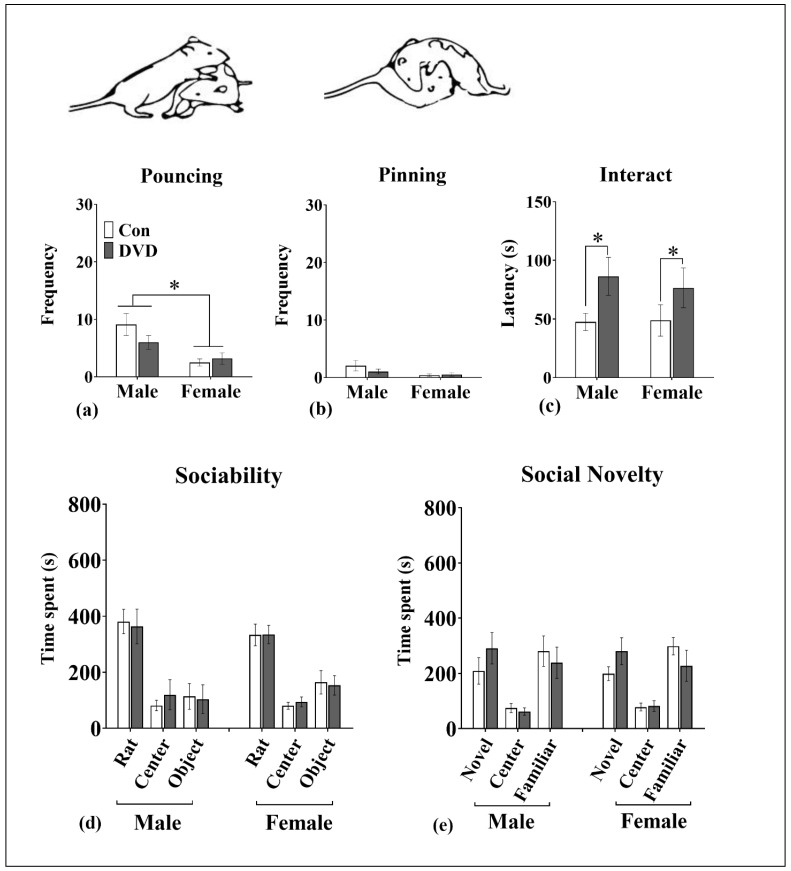
Social play and three chamber social interaction assay (Cohort 3). DVD-deficiency did not affect frequency of (**a**) pouncing and (**b**) pinning. However, DVD-deficiency significantly enhanced (**c**) latency to start interaction in both male and female rats. Male pairs *n* = 24, control male pairs *n* = 24, DVD female pairs *n* = 24 and control female pairs *n* = 22. In the three-chambered assay, (**d**) DVD-deficiency does not affect sociability or (**e**) preference for social novelty. In the sociability phase, experimental rats spent more amount of time in the chamber containing novel rat compared with the novel object. There was also no effect of maternal diet or sex on preference for the rat over object. Social novelty could not be explored as there was no preference displayed for the novel compared to the familiar rat in the control animals. DVD males *n* = 12, control males *n* = 12, DVD females *n* = 12 and control females *n* = 12. Error bars show SEM and * *p* < 0.05.

**Figure 6 nutrients-11-01187-f006:**
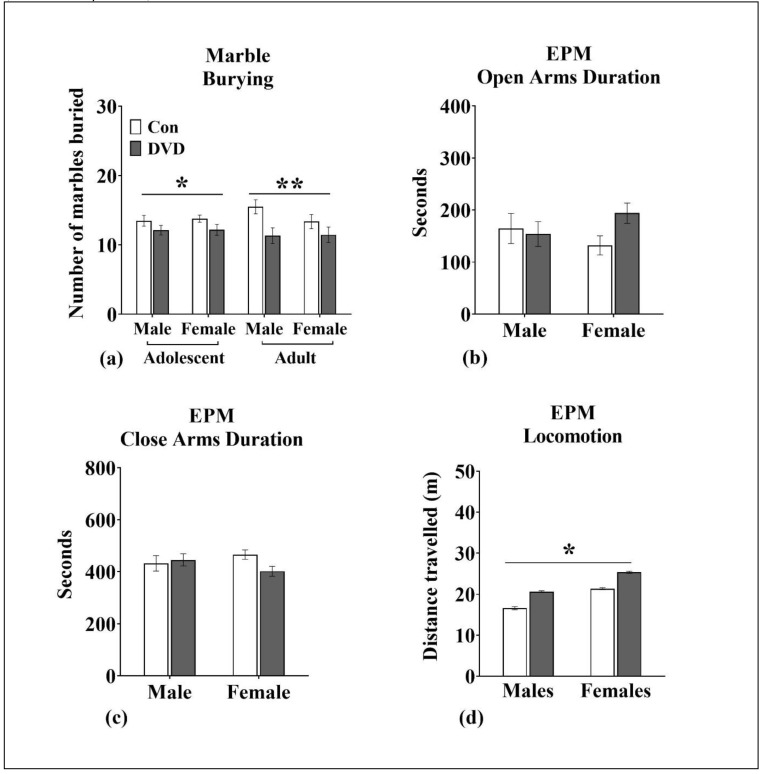
Marble burying and elevated plus maze assay (Cohort 3). (**a**) DVD-deficient rats from both sexes buried significantly fewer marbles than control rats as both adolescents and adults. DVD males *n* = 48, control males *n* = 47, DVD females *n* = 48 and control females *n* = 44. Results show the mean number of marble buried in 30 min of session. In the elevated plus maze. (**b**) DVD-deficiency does not affect the time spent in open arms or (**c**) closed arms. (**d**) Both DVD-deficient male and female rats travelled significantly more than control rats. DVD males *n* = 24, control males *n* = 16, DVD females *n* = 26 and control females *n* = 28. Error bars show SEM, * *p* < 0.05 and ** *p* < 0.01, EPM = elevated plus maze.

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
