# Peer review of "Developmental Vitamin D Deficiency Produces Behavioral Phenotypes of Relevance to Autism in an Animal Model"

_nutrients, 2019, doi:10.3390/nu11051187_

Reviewer 1 Report

The present research study has been focused on important topic such as the vitamin D deficiency in autism rodent models. The study may be considered the starting point for future research in this direction. 

About the study style and organization, the manuscript is well written in each section.

Methods section is well described and rigorous. Findings interpretation is correct. 

Figures are useful for readers and appropriate for contents. 

English is very good and easy to read and References are updated and correct for the topic.

No changes are required because the manuscript doesn't present relevant concerrns. 

Author Response

Point 1: The present research study has been focused on important topic such as the vitamin D deficiency in autism rodent models. The study may be considered the starting point for future research in this direction.

Response: I thank the reviewer for this comment. I agree, this study is just a starting point for future research on the link between gestational vitamin D deficiency and autism. 

Point 2: About the study style and organization, the manuscript is well written in each section.

Response: I thank the reviewer for this comment. 

Point 3: Methods section is well described and rigorous. Findings interpretation is correct. 

Response: I thank the reviewer for this comment. 

Point 4: Figures are useful for readers and appropriate for contents.

Response: I thank the reviewer for this comment. 

 Point 5: English is very good and easy to read and References are updated and correct for the topic.

Response: I thank the reviewer for this comment. 

No changes are required because the manuscript doesn't present relevant concerrns.

Reviewer 2 Report

Well written paper, very relevant. A few comments:

·       Figure 2b – for P9, is there a significant difference? SEM bars are pretty close.

·       Figure 2b – do straight lines which cover many columns mean that there was a difference between the control and treated group within each sex or is the comparison between the sexes (or both)?

·       Is page 10 intentionally mostly blank?

·       In line 427 – the comma after “whether” seems to be unnecessary.

·       Regarding section 4.3 in the discussion: if it has been established that USVs elicit maternal exploratory behavior, with preference to high intensities, I believe the current observation in this paper – lack of sooner retrieval in spite of higher frequencies – should be considered as a finding. In other words – there’s no significant difference where one is expected. Might the DVD-deficient mice possess a quality which alter a maternal instinct? Do the moms know that something’s wrong with the offspring?

Author Response

Point 1: Figure 2b – for P9, is there a significant difference? SEM bars are pretty close.

Response: This graphs shows the difference between the diet groups when animals from both sexes were combined together. The difference becomes significant when animals were pooled for sex in multivariate analysis of variance i.e. p = 0.02.  

Point 2: Figure 2b – do straight lines which cover many columns mean that there was a difference between the control and treated group within each sex or is the comparison between the sexes (or both)?

Response: Straight line shows that the difference is between the diet groups when they were pooled for sex.  

Point 3: Is page 10 intentionally mostly blank?

Response: No it was not intentional. The blank portion on page 10 has been corrected.

Point 4: In line 427 – the comma after “whether” seems to be unnecessary.

Response: The comma on line 427 has been removed.

Point 5: Regarding section 4.3 in the discussion: if it has been established that USVs elicit maternal exploratory behavior, with preference to high intensities, I believe the current observation in this paper – lack of sooner retrieval in spite of higher frequencies – should be considered as a finding. In other words – there’s no significant difference where one is expected. Might the DVD-deficient mice possess a quality which alter a maternal instinct? Do the moms know that something’s wrong with the offspring?

Response: I thank the reviewer for this comment. Reviewer is right. A recent study has shown that DVD-deficiency affect maternal behaviour as they spend less amount of time licking and grooming their pups as compared to controls (Yates et al 2018). DVD-deficient pups were vocalizing more at higher intensities, which may reflect a heightened state of anxiety, so we speculate that dams should behave differently to those pups. However, we have not assessed maternal care behaviour in this study. From the pup retrieval behaviour only, we are unable to conclude if dams know that something’s wrong with the offspring.